# Continuum Modelling for Interacting Coronene Molecules with a Carbon Nanotube

**DOI:** 10.3390/nano10010152

**Published:** 2020-01-15

**Authors:** Kyle Stevens, Thien Tran-Duc, Ngamta Thamwattana, James M. Hill

**Affiliations:** 1School of Mathematical and Physical Sciences, The University of Newcastle, Callaghan, NSW 2308, AustraliaNatalie.Thamwattana@newcastle.edu.au (N.T.); 2School of Information Technology and Mathematical Sciences, University of South Australia, GPO Box 2471, Adelaide, SA 5001, Australia; Jim.Hill@unisa.edu.au

**Keywords:** coronene, carbon nanotubes, stacked columns, continuum modelling, Lennard-Jones potential

## Abstract

The production of single dimensional carbon structures has recently been made easier using carbon nanotubes. We consider here encapsulated coronene molecules, which are flat and circular-shaped polycyclic aromatic hydrocarbons, inside carbon nanotubes. Depending on the radius of the nanotube, certain specific configurations of the coronene molecules can be achieved that give rise to the formation of stacked columns or aid in forming nanoribbons. Due to their symmetrical structure, a coronene molecule may be modelled by three inner circular rings of carbon atoms and one outer circular ring of hydrogen atoms, while the carbon nanotube is modelled as a circular tube. Using the continuous model and the Lennard-Jones potential, we are able to analytically formulate an expression for the potential energy for a coronene dimer and coronene inside a carbon nanotube. Subsequently, stacking of coronene molecules inside a nanotube is investigated. We find that the minimum energy tilt angle of coronenes in a stack differs from that of a single coronene within the same nanotube. More specifically, for both (18, 0) and (19, 0) zigzag carbon nanotube, we find that the minimum energy tilt angles of the single coronene case (≈42${}^{{}^{\circ}}$ and ≈20${}^{{}^{\circ}}$ respectively) do not occur in the stack model.

## 1. Introduction

Clusters of polycyclic aromatic hydrocarbons (PAHs) are of interest to a wide range of fields, from astronomy [1,2] to electrical engineering [3] to environmental science [4,5]. Ordered clusters of PAHs such as crystals [6] and stacked columns [7] have been studied more recently as their production has become increasingly feasible. Stacked columns of PAHs fall under the category of single dimensional nanostructures, a class of materials that are of interest in connection with the development of electronic devices such as supercapacitors [8], transistors [9] and semiconductors [10].

PAHs display similar electronic and thermal behaviour to graphene and are often called nanographenes [11]. Graphene itself has excellent electron transport along its aromatic bonds, though electrons have great difficulty transferring between graphene layers when stacked in graphite [12]. Interestingly, PAHs do not have this issue of interlayer electron transport as electrons are able to travel along π-π stacks quite easily [13,14]. This phenomenon has led to the idea of using these stacks to form nanowires, particularly with larger PAHs such as coronene [15,16]. Rather than long, singular stacks of coronene, these nanowires are clusters of smaller coronene stacks which self organise into fiber-like structures.

In order to combat the inability of coronene to self organise into long stacked columns [17], carbon nanotubes are used to confine the stacking [18,19,20,21,22,23]. Encapsulating coronenes into carbon nanotubes is typically done through vapour-phase doping [18,24]. If a nanotube is too narrow, then coronene molecules cannot enter the tube and if the diameter of the tube is too large then the stacking will become irregular as the molecules can stray off axis [21]. For a window of nanotube diameters, the coronene-nanotube and the π-π interactions restrict the coronene molecules so as to form a single column [23]. There are critical intervals for the diameter for which either the coronene stack will be parallel to the tube’s axis, tilted along the tube’s axis or mostly standing along the tube’s axis [21,23].

Verbeck et al. [22] determined that the diameter of the surrounding nanotube plays a major role in the determination as to whether or not the encapsulated coronene structure will form a columnar stack or anneal to form a graphene nanoribbon. Determining the relationship between the nanotube’s diameter and the encapsulated coronene stack structure is therefore of considerable interest. The primary method of analysis has been the use of simulation techniques including density functional theory (DFT) [18,19], molecular dynamics (MD) [21,23] and Monte Carlo [22]. Okazaki et al. [18] successfully synthesised coronene stacks within a number of zig-zag carbon nanotubes.

To our knowledge, there has been little effort to mathematically model the relationship between the size of the nanotube and the configuration of the coronene stack. An analytical expression that connects the nanotube diameter, the coronene tilt angle and the coronene spacing within the stack might prove extremely beneficial as a predictive tool. To this end, we present a method to mathematically model the stacking of coronene within carbon nanotubes using the continuum approximation together with the Lennard-Jones potential [25]. First we adapt the benzene dimer model developed by Tran-Duc et al. [26] for the coronene dimer and show how to scale this result for any circular PAHs. We then derive a model for a single coronene molecule encapsulated within a carbon nanotube by adapting the model for benzene of Tran-Duc et al. [27]. Thirdly, we present a combination of these two expressions into a scalable model for a coronene stack of any length within a carbon nanotube. We also introduce a variable cut-off distance in order to reduce the number of computations required from this model. Lastly we compare results gained form our model with the current literature and MD simulations.

## 2. Materials and Methods

Coronene’s carbon atoms are arranged radially and hexagonally, and the hydrogen atoms are attached to the outmost carbon atoms. Consequently, a coronene consists of three carbon rings and one hydrogen ring with radii 1.4 Å, 2.8 Å, 3.7 Å, and 4.79 Å, respectively. These radii along with numerical values for constants present in the models are given in Table 1 in the results section. Hence, a coronene molecule can be modelled as four concentric circular rings, three inner rings of carbon atoms and an outer ring of hydrogen atoms as shown in Figure 1. We justify the ring approximation to model the coronene molecule, since it is not uniformly made up of a single element, which would suggest a disc approximation. Rather, the carbon and hydrogen molecules are distributed in such a way that the concentric rings line up perfectly with the atomic coordinates, and approximating the carbon nanotubes as rings is viewed as an unnecessary complication to the model. This approximation approach is similar to a coarse-grained approach which has been succesfully implemented in modelling coronene [28,29].

This approximation allows the coronene-coronene and coronene-nanotube interactions to be expressed as the sums of ring-ring and ring-cylinder interactions. With the aim of deriving an analytical expression for the potential energy of the encapsulated coronene stack, we initially model this from two base interactions. First we model the coronene dimer, which can then be scaled up to a stack model, and then we model a single coronene molecule within a tube. The total interaction energy for the system is then given by
(1)$$E=E_{stack}+\sum E_{coronene-nanotube},$$
where $E_{stack}$ is the interaction energy for the whole stack and $E_{coronene-nanotube}$ is the interaction energy between a coronene molecule and the surrounding nanotube.

For all interaction energies, we derive expressions from the continuum approximation assuming the Lennard-Jones potential between two non-bonded molecules
(2)$$E=\eta_{1}\eta_{2}\int_{S_{1}}\int_{S_{2}}\left(-\frac{A}{\rho^{6}}+\frac{B}{\rho^{12}}\right)dS_{2}dS_{1},$$
where $\eta_{1}$ and $\eta_{2}$ are the surface atomic densities of surfaces $S_{1}$ and $S_{2}$ respectively, *A* and *B* are the attractive and repulsive constants respectively and ρ is the Euclidean distance between two arbitrary points on the two surfaces. We may rewrite Equation (Equation 2) as
(3)$$E=\eta_{1}\eta_{2}\left(-AK_{3}+BK_{6}\right),$$
where $K_{n}\left(n=3,6\right)$ is defined by
(4)$$K_{n}=\int_{S_{1}}\int_{S_{2}}\frac{1}{\rho^{2n}}dS_{2}dS_{1}$$
which is evaluated for n∈Z.

### 2.1. Interaction in a Coronene Dimer

From the assumption that the coronene stack lies coaxially along the nanotube’s central axis, we describe the ring dimer by a displacement along a single axis and a shared tilt angle from the plane normal to this axis (Figure 2). We center one ring at the origin and displace another along the z-axis by a distance δ and we tilt the rings off the xy-plane by an angle ϕ. Due to the symmetry of the problem, we may tilt the rings solely about the y-axis without loss of generality.

Assuming that the rings have two distinct radii *a* and *b*, we can describe an arbitrary point on each ring by (acosϕcosθ,asinθ,−asinϕcos) and (bcosϕcosφ,bsinφ,δ−bsinϕcosφ) for the first and second ring respectively. Using the line elements of each ring, $dS_{1}=ad\theta$ and $dS_{2}=bd\varphi$, we may determine the Lennard-Jones integral between the two rings as
(5)$$K_{n}=ab\int_{0}^{2\pi }\int_{0}^{2\pi }\frac{1}{\rho^{2n}}d\theta d\varphi ,$$
where $\rho =a^{2}+b^{2}+\delta^{2}-2b\delta \sin\phi \cos\varphi +2a\left(\delta \sin\phi -b\cos\varphi \right)\cos\theta -2ab\sin\varphi \sin\theta .$ Equation (Equation 5) is then evaluated and we refer the reader to Appendix A for the main details of this derivation. The final result is as follows
(6)$$\begin{array}{cc} K_{n,\mathrm{rings}} & =4\pi^{2}ab\sum_{i=0}^{\infty }\sum_{j=0}^{i}\sum_{k=0}^{\infty }\frac{{\left(n\right)}_{2i+k}\left(2j\right)!\left(2k\right)!a^{2i}{\left(\delta \sin\phi -b\right)}^{2i-2j}{\left(4b\delta \sin\phi \right)}^{j+k}}{2^{2j+2k}i!j{!}^{2}k{!}^{2}\left(i-j\right)!\left(j+k\right)!{\left(a^{2}+b^{2}+\delta^{2}+2b\delta \sin\phi \right)}^{n+2i+k}}, \end{array}$$
which, although clearly very complicated, may be readily evaluated using standard algebraic packages such as MAPLE and MATLAB. This is a special case of the model given by Tran-Duc et al. [26] in which involves a more general displacement and rotation of the second ring. The present model is more useful for the purpose of extending to a stack as superfluous variables may be disregarded.

Assuming the four ring approximation of coronene, we set up a coronene dimer as four concentric rings displaced vertically from four more concentric rings (Figure 3a). We need to sum up the pairwise interactions that each ring experiences, amounting in total to sixteen pairwise energies, in order to calculate the total interaction energy.

More generally, for two molecules with $\nu_{1}$ and $\nu_{2}$ concentric rings respectively we have $\nu_{1}\times \nu_{2}$ interactions to consider, to give a total energy
(7)$$E_{\mathrm{dimer}}=\sum_{i=1}^{\nu_{1}}\sum_{j=1}^{\nu_{2}}E_{ij}\left(\delta ,\phi \right),$$
where $E_{ij}$ is the interaction energy between the *i*th ring of the first molecule and the *j*th ring of the second molecule respectively.

### 2.2. Interaction between a Carbon Nanotube and an Encapsulated Coronene

A circular ring of radius *a* is centered at the origin and rotated about the y-axis by an angle ϕ. An infinitely long tube of radius *c* is placed such that its axis coincides with the z-axis. Let a<c, so that the ring is located within the tube (Figure 4).

An arbitrary point on either surface has coordinates (acosϕcosθ,asinθ,−asinϕcosθ) and (bcosφ,bsinφ,z) on the ring and tube respectively. With the two surface elements $dS_{ring}=a^{2}d\theta$ and $dS_{tube}=cdzd\varphi$, we can then set up the integral $K_{n}$ as
(8)$$\begin{array}{cc} K_{n} & =a^{2}c\int_{0}^{2\pi }\int_{0}^{2\pi }\int_{-\infty }^{\infty }\frac{1}{\rho^{2n}}dzd\varphi d\theta , \end{array}$$
where $\rho ={\left(a\cos\phi \cos\theta -b\cos\varphi \right)}^{2}+{\left(a\sin\theta -b\sin\varphi \right)}^{2}+{\left(z+a\sin\phi \cos\theta \right)}^{2}$. An analytical evaluation of Equation (Equation 8), the full details of which can be found in Appendix B, is given by
(9)$$\begin{array}{cc} K_{n,\mathrm{tube}} & =\frac{\pi^{3}a\left(2n-3\right)!!}{2^{n-3}\left(n-1\right)!c^{2n-2}}\sum_{i=0}^{\infty }\sum_{j=0}^{i}\frac{{\left(n-1/2\right)}_{i}^{2}{\left(-\sin^{2}\phi \right)}^{j}\left(2j-1\right)!!}{2^{j}i!{\left(j!\right)}^{2}\left(i-j\right)!}{\left(\frac{a}{c}\right)}^{2i}, \end{array}$$
which is a restricted case of the model derived by Tran-Duc et al. [27].

As with the dimer model, we again consider a coronene molecule as four concentric rings, each of which is centered at the origin within the tube, as shown in Figure 5a.

Summing each pairwise interaction between the rings and tube gives the total potential energy as a function of the tube radius and tilt angle,
(10)$$E_{\mathrm{total}\;\mathrm{tube}}=\sum_{i=1}^{4}E_{\mathrm{tube}}\left(c,\phi ;a_{i}\right),$$
where $a_{i}$ is the radius of the *i*th ring in the coronene approximation.

### 2.3. Interaction between a Carbon Nanotube and an Encapsulated Coronene Stack

Firstly we need to consider a stack of coronene molecules, each of which is approximated as four concentric rings, that all have the same tilt angle ϕ.

To simplify the calculation, we assume that the molecules are all equidistant along the axis and separated by δ, as shown in Figure 6a. This allows for the total energy of the stack to be calculated by summing all of the dimer interactions present, though a cut off distance is needed to keep the computation time down when the stack is arbitrarily large. The cut off distance varies depending on the atoms present in the interaction and is usually taken to be 2.5σ where σ is the distance at which the interaction energy is zero [30]. We have allowed for a variable cut off distance based on multiples of δ; for a cutoff of *n* molecules and a stack of *N* molecules, we need a total of $\sum_{k=1}^{n}\left(N-k\right)=nN-n\left(n+1\right)/2$ dimer interactions to be calculated. The total energy of the entire stack is therefore given as
(11)$$\sum_{i=1}^{n}\sum_{j=1}^{N-i}E_{\mathrm{dimer}}\left(i\delta ,\phi \right),$$
where iδ is the distance between two molecules that are *i* molecules apart in the stack. If the cut off is larger than the total number of molecules, then we alter Equation (Equation 11) to obtain
(12)$$\sum_{i=1}^{N-1}\sum_{j=1}^{N-i}E_{\mathrm{dimer}}\left(i\delta ,\phi \right).$$

Considering the stack within a tube ( Figure 7a) we add the interaction energy between each coronene molecule and the nanotube, which based on our assumptions are each identical. This gives the total energy for the entire system of a coronene stack within a nanotube as
(13)$$\sum_{i=1}^{n}\sum_{j=1}^{N-i}E_{\mathrm{dimer}}\left(i\delta ,\phi \right)+N\cdot E_{\mathrm{total}\;\mathrm{tube}}.$$

## 3. Results and Discussion

### 3.1. Interaction in a Coronene Dimer

We aim to find the value of δ for which the dimer energy is minimised at various angles of interest. By fixing ϕ, we can find the displacement δ by solving ∂E/∂δ=0. We see in Figure 8 that as ϕ increases, the displacement at which the minimum energy occurs increases, and that the minimum energy increases as the tilt angle increases.

Dappe et al. [19] determined that for zero tilt angle the displacement between the coronenes lies between 3.1 and 3.2 Å with potential energies between −1.3 and −1
eV (≈−30 to $-23\;\mathrm{kcal}.{\mathrm{mol}}^{-1}$). The displacement and energy values are close to those derived values in Table 2. Values for $30^{{}^{\circ}}$ (≈ 3.65 Å and $-25\;\mathrm{kcal}.{\mathrm{mol}}^{-1}$) and $45^{{}^{\circ}}$ (≈ 4.5 Å and $-19.5\;\mathrm{kcal}.{\mathrm{mol}}^{-1}$) are also in agreement with values predicted here. Dappe et al. [19] also note that the relative angle about the shared axis between two coronene molecules in a dimer also makes a difference in potential energy. This effect is not possible to capture using the continuum approximation as it is largely a discrete effect arising from the position of the atoms within the molecules.

### 3.2. Interaction between a Carbon Nanotube and an Encapsulated Coronene

Here we are concerned primarily with finding a range of tube radii where the coronene molecule will be tilted. Due to the symmetry of the model we find that, for any tube radius, ϕ=0 and ϕ=π/2 are always stationary points. We may compute ${\left.\partial^{2}E/\partial \phi^{2}\right|}_{\phi =0}=0$ and ${\left.\partial^{2}E/\partial \phi^{2}\right|}_{\phi =\pi /2}=0$ to determine where the concavity change occurs, which indicates a change in regime from lying to tilted to standing. The tube radii favouring a lying coronene molecule are found to be for b≤
6.8395 Å while for b≥
7.6411 Å has the coronene standing on axis. This gives a small window for radii for which the coronene lies tilted on the axis of the nanotube. These intervals can be seen in Figure 9a, and a closer view for tube radii within this interval is shown in Figure 9b.

The relationship between the tube radius and the optimal tilt angle of the coronene is shown in Figure 10 to be non-linear within the tilted regime.

From Table 3 we see that only two zig-zag carbon nanotubes cause an encapsulated coronene molecule to be tilted. Anoshkin et al. [31] use DFT simulations to determine the tilt angle and the potential energy of a single coronene molecule inside three different carbon nanotubes: (15,0), (18,0) and (20,0). They determine that for the (15,0) tube, there is no energetically favourable tilt angle. For the (18,0) tube, they determine the tilt angle of $60^{{}^{\circ}}$ from the tube’s axis with a potential energy of −1.16
eV (≈ −27
kcal
${\mathrm{mol}}^{-1}$). The (20,0) tube induces a tilt angle perpendicular to the tube’s axis, with potential energy of −1.1
eV (≈ −25
kcal
${\mathrm{mol}}^{-1}$).

In comparison with their results, equilibrium energy values in the current study are overestimates, while the tilt angles give better agreement. This discrepancy in the energy calculation might arise from discrete effects that are not incorporated in a continuum approximation. The distances between the coronene molecule and the nanotube are around 2− 3 Å, which are distances where the Lennard-Jones potential is typically very repulsive. This means that adding atoms where technically there are none, which the continuum approach does, will alter the energy profile and typically increase repulsive effects.

In order to explore the discrete effect, a molecular dynamic (MD) simulation is carried out using LAMMPS package. The interaction energy between the coronene and the carbon nanotube is the summation of pairwise interactions between atoms of the two molecules using the discrete Lennard-Jones potential as the force field. In the simulation, the coronene molecule is rotated about its own axis and the energy is then recorded. The discrete effects can be seen clearly in Figure 11, particularly in Figure 11a, where the potential energy at a particular tilt angle also varies with the coronene rotation about its own axis.

### 3.3. Interaction between a Carbon Nanotube and an Encapsulated Coronene Stack

Okazaki et al. [18] observed experimentally that, in a (19,0) nanotube, a coronene stack will have an optimal tilt angle of $77^{{}^{\circ}}$ from the tube’s axis, with a spacing of about 3.5 Å. This corresponds closely to the minimum energy value found in Table 4. They also note that the optimal angle has contributions from both the coronene-coronene and coronene-nanotube interactions, which is also shown in Table 4 and Table 5. We show this by taking the optimal angle from the single coronene case (the center value in each table) and then use this value to fix the angle for the stack case, from which we calculate the spacing distance. We then vary this fixed angle by small amounts (π/100,π/50 and π/25) either side to see how it affects the spacing and the potential energy.

Sakane et al. [21] also found, using molecular dynamics simulations and conjugate gradient minimisation, that the radius of the nanotube encapsulating a coronene stack is the main cause of the mean tilt angle of the stack until it becomes large (≈ 7.6 Å). From their results, nanotubes of similar radii to (18,0) and (19,0) tubes exhibit mean tilt angles of ≈42${}^{{}^{\circ}}$ and ≈30${}^{{}^{\circ}}$ respectively, with mean coronene spacing of ≈ 4.51 Å and ≈ 3.87 Å. This agrees reasonably well with the results listed in both Table 4 and Table 5 here.

Kigure et al. [32] determine that the tilt angle of coronene molecules stacked within a carbon nanotube affect the electronic structure of the stack. This is due to the hybridisation of the π-π interaction between each coronene molecule which increases as the tilt angle increases away from the nanotube’s axis. Thus, being able to predict the tilt angle of a coronene stack is crucial when aiming to produce a stack of specific electronic configuration.

## 4. Conclusions

In this paper, we use a continuum approach together with the Lennard-Jones potential function to model the stacking of coronene molecules within carbon nanotubes. Analytical expressions for the interaction energy in coronene dimer and between a coronene molecule encapsulated within a carbon nanotube are obtained. These are then combined in such a way that we obtain an analytical expression for the interaction energy of a stack of coronene molecules within a carbon nanotube. We find that, under our assumptions, a coronene dimer has its least energy configuration at a tilt angle of zero and a displacement of 3.2651 Å. We then investigate a single coronene molecule encapsulated within several zig-zag carbon nanotubes and find that only (18,0) and (19,0) nanotubes induce a tilt angle between zero and π/2. Moreover, we find that there is only a small window of nanotube radii for which a coronene molecule will be tilted within the carbon nanotube. Lastly we examine the tilt angle ϕ of a coronene stack and find it to depend on both the carbon nanotube radius and the displacement between coronene molecules δ within the stack. Interestingly, the tilt angle of a coronene stack within a carbon nanotube is found to be less than that of a single coronene molecule within the same carbon nanotube.

We comment that while modern facilities permit MD simulations of the same force field at a fully atomistic level, as well as even more general problems, the detailed mathematical analysis presented here represents the limit of what might be achieved using the continuum modelling approach. We also comment that the modelling adopted here might be viewed as a special case of a much more general problem in which the coronene could be described by either a set of rings or by a uniform disc, and the same two approximations could be used for the nanotube that might be compared with a full atomistic MD simulation, and this is a potential subject for future investigation.

## Figures and Tables

**Figure 1 nanomaterials-10-00152-f001:**
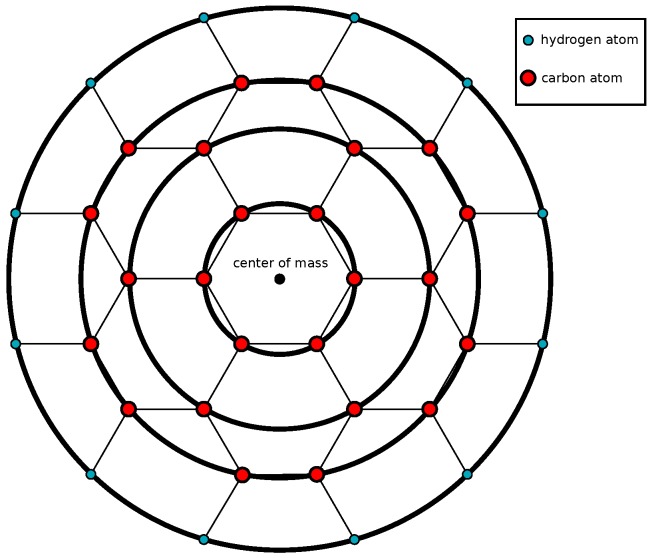
Model of coronene molecule with inner carbon rings and outer hydrogren ring.

**Figure 2 nanomaterials-10-00152-f002:**
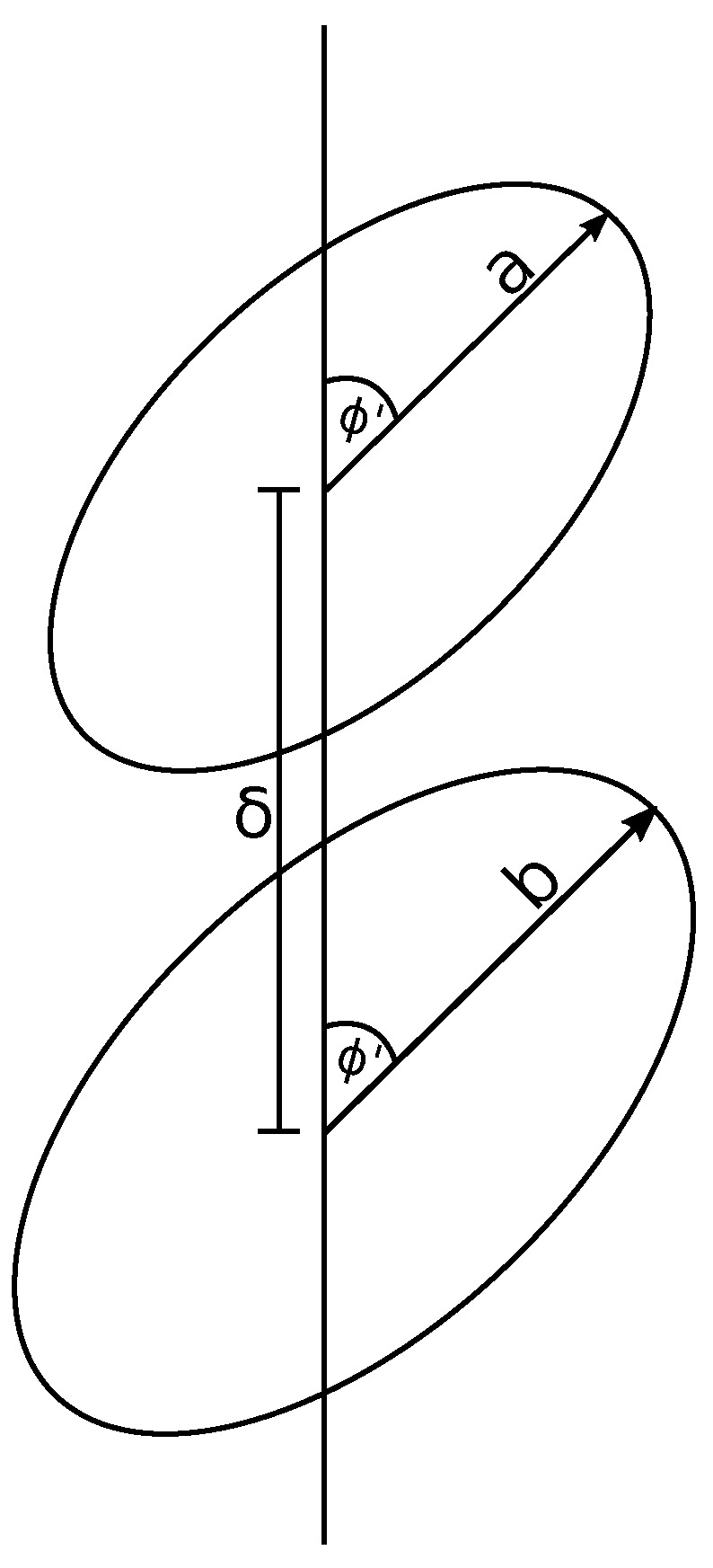
Two rings displaced by δ and both tilted by the same angle ϕ.

**Figure 3 nanomaterials-10-00152-f003:**
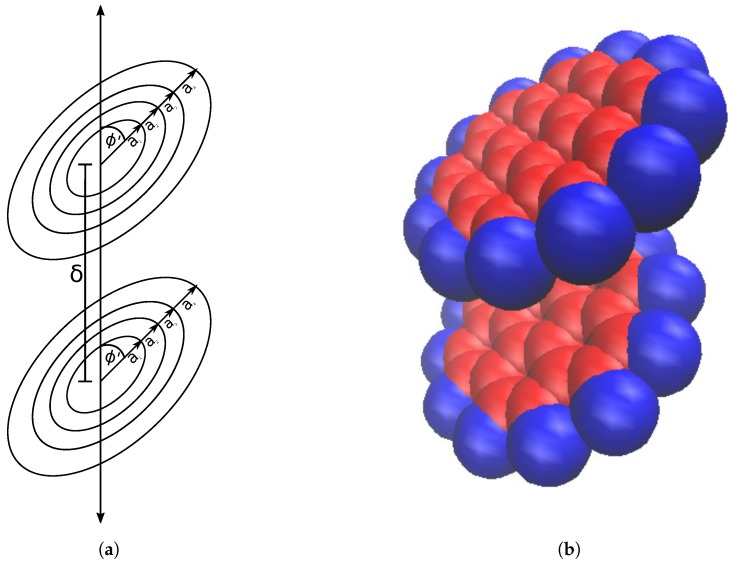
Coronene dimer represented by (**a**) concentric circular rings and (**b**) space filling model

**Figure 4 nanomaterials-10-00152-f004:**
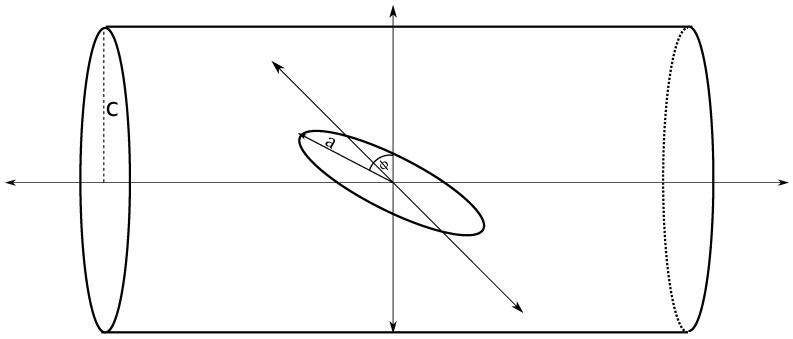
Circular ring lying within tube and centered on tube’s central axis and tilted by angle ϕ.

**Figure 5 nanomaterials-10-00152-f005:**
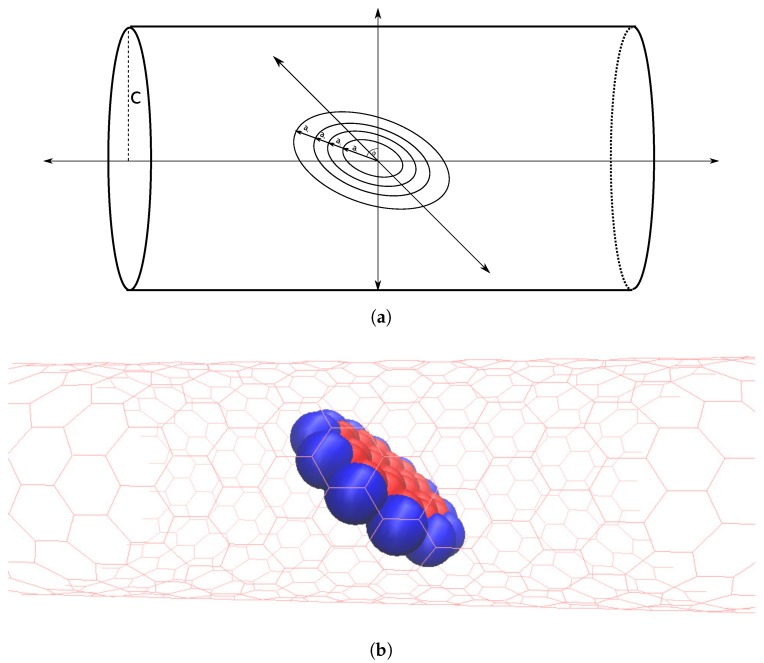
Coronene molecule within tube (**a**) approximated by four concentric rings; centered on tube’s central axis and tilted by an angle ϕ and (**b**) represented by space filling model

**Figure 6 nanomaterials-10-00152-f006:**
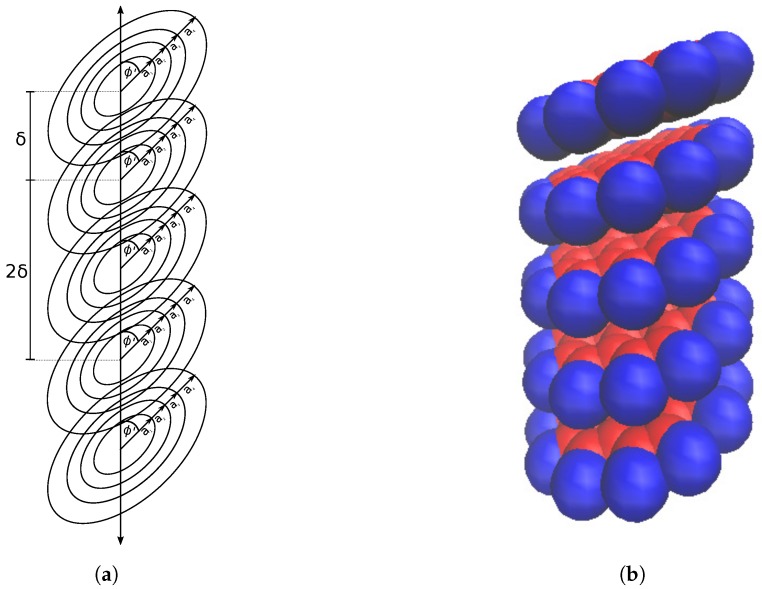
Stack of coronene molecules (**a**) approximated by concentric rings; all with equidistant placing along axis by distance δ and all tilted by the same angle $\phi^{\prime }=\pi /2-\phi$ and (**b**) represented by space filling model.

**Figure 7 nanomaterials-10-00152-f007:**
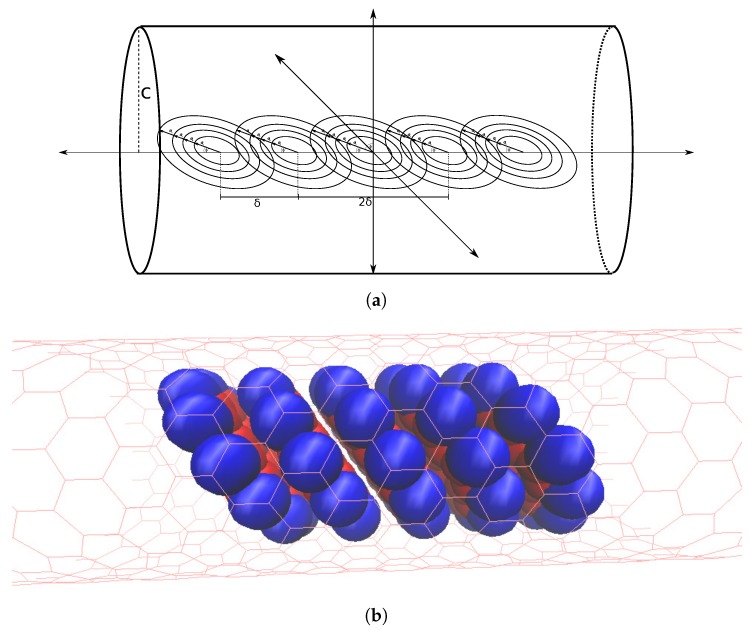
Stack of coronene molecules (**a**) approximated by concentric rings, all equidistant along central axis of nanotube, by distance δ and all tilted by same angle ϕ and (**b**) represented by space filling model.

**Figure 8 nanomaterials-10-00152-f008:**
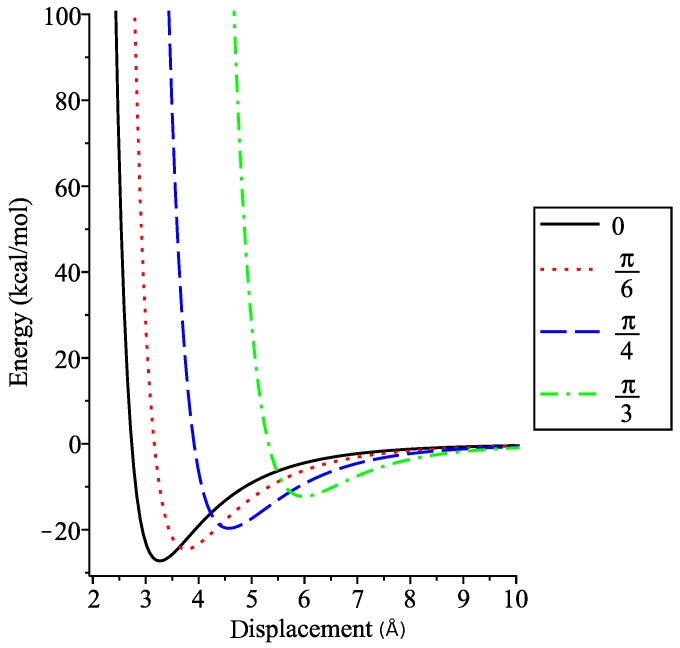
Interaction energy within coronene dimer for fixed angles.

**Figure 9 nanomaterials-10-00152-f009:**
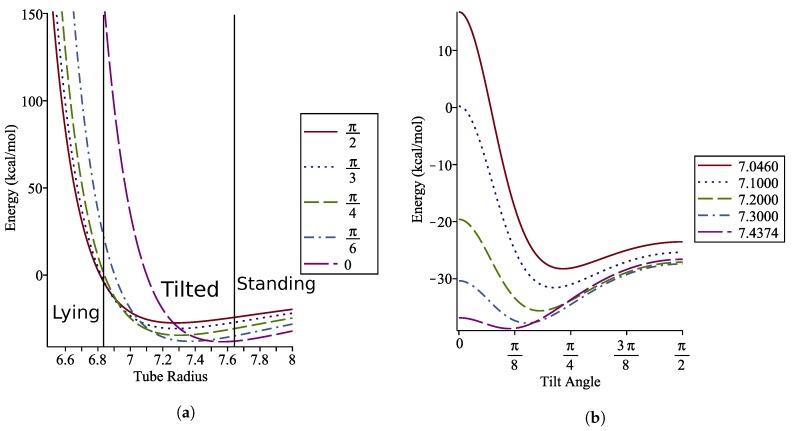
Energy profiles with (**a**) fixed angles showing the minimum energy regimes and (**b**) fixed radii showing variation in optimal angles in tilted regime

**Figure 10 nanomaterials-10-00152-f010:**
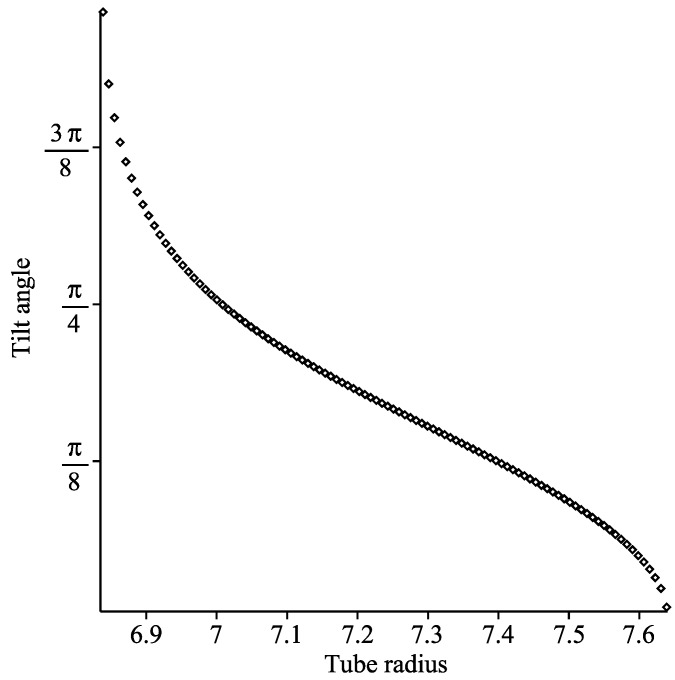
Graph showing optimal tilt angle of coronene molecule as tube radius varies.

**Figure 11 nanomaterials-10-00152-f011:**
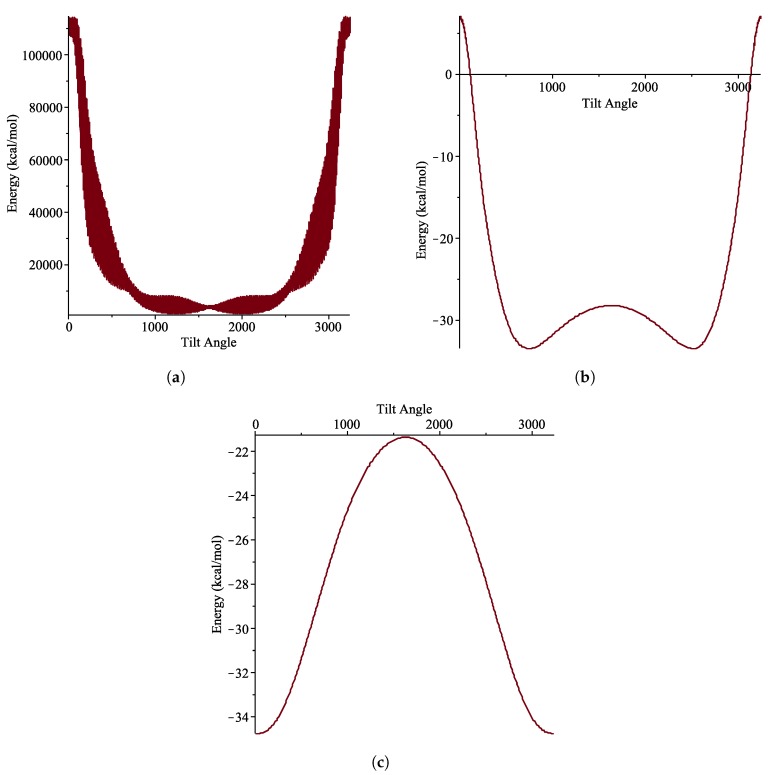
Energy distributions for given tilt angle (in π/3240) where coronene is rotated about own axis as well as y-axis for (**a**) (15, 0) tube (**b**) (18, 0) and (**c**) (20, 0).

**Table 1 nanomaterials-10-00152-t001:** Numerical values of constants used in present model.

First carbon ring radius	1.400 Å
Second carbon ring radius	2.800 Å
Third carbon ring radius	3.702 Å
Hydrogen ring radius	4.793 Å
First carbon ring atomic line density	0.682 Å^−1^
Second carbon ring atomic line density	0.341 Å^−1^
Third carbon ring atomic line density	0.516 Å^−1^
Hydrogen ring atomic line density	0.398 Å^−1^
CNT atomic surface density	0.382 Å^−2^
CNT (16,0) radius	6.263 Å
CNT (17,0) radius	6.655 Å
CNT (18,0) radius	7.046 Å
CNT (19,0) radius	7.437 Å
CNT (20,0) radius	7.829 Å
C-C attractive constant	560.44kcal${\mathrm{mol}}^{-1}$ Å^6^
C-C repulsive constant	1121755.66kcal${\mathrm{mol}}^{-1}$ Å^12^
C-H attractive constant	129.67kcal${\mathrm{mol}}^{-1}$ Å^6^
C-H repulsive constant	91727.95kcal${\mathrm{mol}}^{-1}$ Å^12^

**Table 2 nanomaterials-10-00152-t002:** Minimum displacements and corresponding interaction energies for fixed tilt angles.

Angle (radians)	Displacement (Å)	Energy ($\mathrm{kcal}\;{\mathrm{mol}}^{-1}$)
0	3.2651	−27.1774
π/6	3.7603	−24.4359
π/4	4.5739	−19.5415
π/3	5.9861	−12.1282

**Table 3 nanomaterials-10-00152-t003:** Optimal tilt angles and corresponding potential energies for single coronene molecule within carbon nanotube of specified radius.

Nanotube (n,m)	Angle (radians)	Energy ($\mathrm{kcal}\;{\mathrm{mol}}^{-1}$)
(17,0)	π/2	50.7784
(18,0)	0.7318	−28.2533
(19,0)	0.3543	−38.7317
(20,0)	0	−35.3366

**Table 4 nanomaterials-10-00152-t004:** Minimum energy intermolecular spacings corresponding to given angles for stack of three coronene encapsulated within (19,0) nanotube.

Angle (radians)	Coronene Spacing (Å)	Energy ($\mathrm{kcal}\;{\mathrm{mol}}^{-1}$)
0.2543	3.3399	−170.6357
0.2943	3.3765	−170.7773
0.3243	3.4084	−170.6263
0.3543	3.4442	−170.2269
0.3843	3.4842	−169.5614
0.4143	3.5295	−168.6260
0.4543	3.5969	−166.9631

**Table 5 nanomaterials-10-00152-t005:** Minimum energy intermolecular spacings corresponding to given angles for stack of three coronene encapsulated within (18,0) nanotube.

Angle (radians)	Coronene Spacing (Å)	Energy ($\mathrm{kcal}\;{\mathrm{mol}}^{-1}$)
0.6318	4.0143	−131.2193
0.6718	4.1378	−130.6053
0.7018	4.2383	−129.6444
0.7318	4.3458	−128.3193
0.7618	4.4608	−126.6789
0.7918	4.5839	−124.7670
0.8318	4.7613	−121.8610

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
