# Peer review of "Continuum Modelling for Interacting Coronene Molecules with a Carbon Nanotube"

_nanomaterials, 2020, doi:10.3390/nano10010152_

Round 1
Reviewer 1 Report
The manuscript by Stevens et al. reports the derivation and a approbation of analytical expression within a continuum model for interaction between two disk-shaped molecules and between disk-shaped molecule and cylindrical nanotube. The Authors explore the orientation of a stack of coronene molecules in endohedral (within nanotube) state. A relation between the nanotube diameter and tilting angle of endohedral coronenes is established and is approved using fully atomisic calculations as well using the literature data. In my opinion this manuscript may be published in the current version.
The only minor correction could be made at lines 62-63. The Authors wrote: "...a coronene consists of three carbon layers and one 63 hydrogen layer..." I cannot distinguish "layers" in coronene. It is better to replace "layers" on the "rings" immediately.
As well, I find this study quite astonishing, since modern facilities permit without problems the MD simulations within the same force-field, but at fully atomistic level. In contrast to continuum model, a fully atomistic model generalize the problem easily and is capable to include the symmetry lowering: a deviation of coronene barycenter from the tube axis could be easily considered as well as the hexagonal symmetry of coronene could be accounted.
Author Response
We have replaced the mention of layers with the use of the term rings (page 2, line 64).
We have added some lines in the conclusion relating to an atomistic MD approach or the use of the continuum model (page 14, line 224).
Reviewer 2 Report
Review of “Continuum Modelling for Interacting Coronene Molecules with a Carbon Nanotube by K. Stevens, T. Tran-Duc, N. Thamwattana and J. M. Hill”
This is a very nice paper which describes mathematical modelling based on both Continuum Modelling and Lennard-Jones potentials both how coronene molecules can stack inside carbon nanotubes, including both horizontal stacking and angled stacking. The paper is very well written and there are few or no typographical issues.
Regarding the modelling, I have only a few comments and suggestions. Can the authors make clearer in their analysis how the composite structures ‘fit together’ ? Elsewhere, this has been done by taking into account the inner surface of the carbon nanotubes (CNTs) in terms of their diameter but also the van Der Waals radii of the constituent atoms related to the outside volume of the coronene molecules which also in turn determines the stacking both horizontally but also tilted. Of course this is accounted for in the Lennard Jones potential but composite space filling models in Figures 5, 6 and 7 would give a clearer impression of where the horizontal stacking limit is imposed by the inner surface of the CNTs. This can be extrapolated from Table 1 but space filling models based on the radii in this table would give the non-specialist a better feel for the stacking versus the ball and stick representations given in Figures 3(b), 5(b), 6(b) and 7(b) in particular. Other than this, I can recommend this article for publication in Nanomaterials.
Author Response
We have replaced the ball and stick models in Figures 3(b), 5(b), 6(b) and 7(b) with space filling models (see pages 5, 6, 7 and 8).
Reviewer 3 Report
Referee report on Nanomaterials-666192
Continuum modelling for coronene molecules and carbon nanotube
by K. Stevens, T. Tran-Duc, N. Thamwattana, J. Hill
The authors have derived the exact calculation of the interaction
energy between a single or a stacked set of coronene molecules
inserted into a carbon nanotubes, using continuum type approximations
for the coronene molecules (as four rings) or the tube itself (uniform
cylinder). Energetic and structural details for the optimal angles in
the stack are given for different nanotube radii.
This work attempts a rather coarse-grained approach at the interaction
between hydrocarbon compounds, following similar ideas as recently
laid out by other groups (J Chem Phys 143, 174110, 2015; Phys Chem
Chem Phys 18, 13736, 2016).
Here the work might appear a bit academic and would appear more
convincing if the authors could compare (favorably) with atomistic
calculations employing the same LJ parameters.
Also, how is the ring approximation justified for coronene, and why
not using it for some zigzag nanotubes?
I see the work conducted by the authors as a special case of a much
more general problem, in which the coronene could be described by
either a set of rings or by a uniform disc, and the same two
approximations could be used for the nanotube. Both being compared to
all-atom predictions, or even to intermediate approximations such as
an homogeneous nanotube but an atomistic coronene (analytical
expressions also available for this problem).
I encourage the authors to expand their work and at least provide some
comparison with atomic simulation data (optimal structures) and show
to which extent their analytical model is realistic and quantitative.
Author Response
We have noted the similarities to the coarse-grained approach (page 2, line 70).
We have compared our model’s results to DFT simulation results or other MD simulations found in literature (page 13, line 198). To our knowledge, we have not found any atomistic LJ simulations in literature.
We have incorporated the following sentences in the manuscript (page 2, line 65).
“We justify the ring approximation to model the coronene molecule, since it is not uniformly made up of a single element, which would suggest a disc approximation. Rather, the carbon and hydrogen molecules are distributed in such a way that the concentric rings line up perfectly with the atomic coordinates, and approximating the carbon nanotubes as rings is viewed as an unnecessary complication to the model.”
We have noted that our future work will include atomistic LJ simulations with which we can directly compare our model (page 14, line 230). We have also noted that in future work we may consider the more general case of modelling the nanotubes as rings of carbon as opposed to cylinders of carbon (page 14, line 229).
Round 2
Reviewer 3 Report
I am overall satisfied with this revision.
I understand and agree with the 'unnecessary' complication that would be associated with treating the nanotube as an infinite series of rings (but, again, I would be curious to know which of the discrete/continuum approximation is more important between the two interacting carbonaceous structures).
Concerning my remark related to the needed comparison with atomistic calculations, I was simply hoping the authors could do it themselves, I would not expect thousands of line codes there since it is only the LJ potential after all.
I recommend this revision for publication in Nanomaterials;